# Collective intelligence: A unifying concept for integrating biology across scales and substrates
Patrick McMillen[1,2] & Michael Levin [1,2,3] ✉

A defining feature of biology is the use of a multiscale architecture, ranging from molecular networks to cells, tissues, organs, whole bodies, and swarms. Crucially however, biology is not only nested structurally, but also functionally: each level is able to solve problems in distinct problem spaces, such as physiological, morphological, and behavioral state space. Percolating adaptive functionality from one level of competent subunits to a higher functional level of organization requires collective dynamics: multiple components must work together to achieve specific outcomes. Here we overview a number of biological examples at different scales which highlight the ability of cellular material to make decisions that implement cooperation toward specific homeodynamic endpoints, and implement collective intelligence by solving problems at the cell, tissue, and whole-organism levels. We explore the hypothesis that collective intelligence is not only the province of groups of animals, and that an important symmetry exists between the behavioral science of swarms and the competencies of cells and other biological systems at different scales. We then briefly outline the implications of this approach, and the possible impact of tools from the field of diverse intelligence for regenerative medicine and synthetic bioengineering.

One defining feature of complex life, making it distinct from our current engineered artifacts, is its multiscale nature: there is order in biology across levels of organization, from molecules to cells, tissues, organs, whole organisms, and societies/swarms[1,2]. Crucially, however, this goes well beyond structural nesting: it is in fact a multiscale competency architecture[3,4] because each level solves problems in its own relevant domains (Fig. 1). As evolution facilitated the increase of complexity, living things became composed of layers that cooperate and compete to solve problems in metabolic, physiological, anatomical, and behavioral state spaces (reviewed in refs. [5,6]). Biology's robustness, open-endedness, evolvability, and unique complexity likely depend on the fact that evolution works with an agential material – a substrate with competencies, computational abilities, and homeodynamic setpoints[5,7] that strongly influence the structure and function of multicellular forms. Adaptive behavior in new problem spaces[3,4] can arise because higher levels of organization can deform the energy landscape for the subunits[8], while benefitting from their ability to navigate those landscapes autonomously and without micromanagement.

Understanding how the behavior of subunits percolates up toward adaptive processes at higher levels (Fig. 1a–e), and how higher levels of organization constrain and facilitate the behavior of their parts[9–15], is critical not only to basic evolutionary biology but also to the control of system-level outcomes in biomedicine[16,17] and to the design of novel engineered systems[18–24]. We have previously proposed that this research program can be advanced by exploiting collective intelligence as a crucial symmetry across levels, which enables the tools of behavioral science to be brought to bear on novel unconventional substrates[16,17,25,26], especially the capabilities of cell groups in transcriptional, physiological, and anatomical spaces (Fig. 1c, e). Specifically, we have argued that regulative morphogenesis is a kind of behavior of cellular collectives traversing anatomical morphospace (Fig. 1f, g)[27–32], and others have argued that immune systems[33,34], bacterial biofilms[35–38], and many other unconventional substrates[39–42] can be effectively understood and rationally controlled by using techniques from behavioral and cognitive science[43].

Here, we explore a number of phenomena in biology which illustrate this approach, specifically focusing on two aspects that feature prominently in behavioral science. One is *intelligence*, in William James'[44] sense of a degree of ability to reach the same goal by different means (i.e., problem-solving in changing or novel circumstances). The other is collective

[1]Department of Biology, Tufts University, Medford, MA 02155, USA. [2]Allen Discovery Center at Tufts University, Medford, MA 02155, USA. [3]Wyss Institute for Biologically Inspired Engineering, Harvard University, Boston, MA 02115, USA. ✉e-mail: michael.levin@tufts.edu

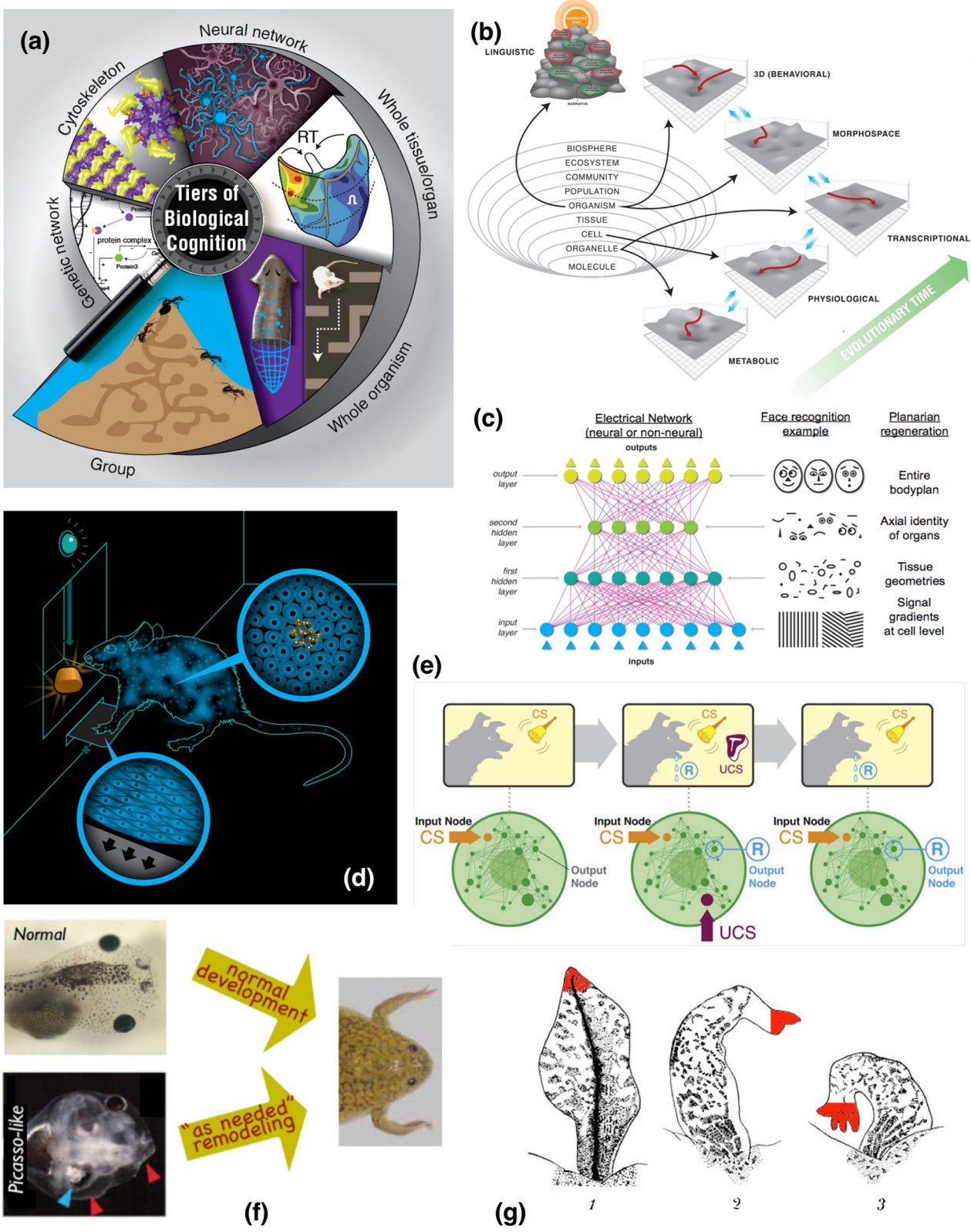

decision-making, as studied in the rapidly advancing study of group behavior among swarms[45–50]. This parallel has been explored previously[51–54], and we extend those ideas here with specific references to more recent data revealed by advances in non-invasive imaging and functional cell modulation technology. We emphasize organizational principles that enable not just emergent complexity, but adaptive proto-cognitive systems (problem-

solving with respect to adaptive goals and novel circumstances) to appear[3,55]. A central claim of the emerging field of diverse intelligence is that cognitive capacities (Box. 1) exist on a spectrum: that tools, concepts, and approaches from behavioral sciences can be productively applied to understand and control systems far beyond familiar animals with central nervous systems (without the necessity to attribute advanced, human-level metacognitive

**Fig. 1 | Wholes and parts in biological systems: collective intelligence of competent parts is projected into novel problem spaces by higher-scale systems.**
**a** Living bodies implement a multiscale competency architecture in which each level of organization, from molecular networks to swarms of animals, navigates specific problem spaces, such as metabolic, physiological, transcriptional, anatomical, and behavioral landscapes. As these diverse subsystems cooperate and compete with each other, their problem-solving dynamics constate adaptive collective intelligence. **b** Indeed, one way to view evolution of complex forms is as a re-use of many of the same mechanisms and strategies across scales of organization[7,72] and problem spaces, which may even extend to high-level navigation of linguistic space. **c** Concepts from connectionist machine learning, such as artificial neural networks, now provide a rigorous, quantitative understanding of ways in which higher-level information is derived from lower-level subsystems' inputs in a collective system: for example, input layers receive pixel-level information, but each subsequent layer extracts progressively higher level features in the image[8]. **d** The percolation of information across scales is a fundamental aspect of neuroscience: a rat which has learned to press a lever to get a reward is an emergent collective agent, consisting of large numbers of cells, none of which had both experiences on the relevant timescale (interacting with the lever or receiving the nutrients). The cognitive glue that enables emergent agents to support associative memories over their subunits is neural bioelectricity in the case of conventional 3D world behavior, as well as in the traversal

of morphospace during development or regeneration[27]. **e** Even subcellular components are likely to participate in the scaling of emergent entities from competent parts, as networks as simple as small gene-regulatory circuits or pathways can support several different kinds of learning, including Pavlovian conditioning, when the individual nodes participate in time-dependent stimulus and response patterns[225–227]. **f** At the level of tissues and organs, collective problem-solving is observed in phenomena such as regulative metamorphosis, in which tadpoles with incorrect arrangements of craniofacial organs still become normal frogs, by novel movements of entire complex structures that operate to reduce distance (error) from current configuration to the normal frog target morphology. This system represents an ideal example of William James' definition of intelligence as a capacity to achieve specific ends by diverse means as necessary. **g** Other powerful but poorly-understood examples of collective decision-making include the progressive transformation of a tail transplanted to the flank of an amphibian into a limb: the distal cells (in red)[228] slowly become toes, even though in their *local* environment nothing is wrong (tail tip cells located at the tip of the tail): it is a collective decision that transforms them, flowing down from a perception of anatomical error that is only defined at the whole body-level. Panels a-e created by Jeremy Guay of Peregrine Creative, and used by permission from Refs. [3,5,25,27,225] respectively. Panels f and g are taken with permission from Ref. [151]. and[228] respectively.

**Fig. 2 | Collective Intelligence expands the perceptual field of a group of cells beyond the capacity of any individual, expanding their problem solving ability.** The competency of subunits that flows upward to the collective decision-making is a kind of intelligence – the ability to navigate an environment in an adaptive way that enables specific homeostatic or homeodynamic goals to be met despite novel perturbations or barriers. Non-living objects are capable of simple goal directed behavior, as in the case of a bar magnet moving towards its counterpart (**a**). This movement can be blocked by an impediment in the direct path of the magnet. Intelligent systems also exhibit goal directedness with the addition of continually surveying their environment in both time and space to find alternate paths to achieve their goal (b). Biological intelligent systems demonstrate increased ability to achieve their (collective) goals despite obstacles by integrating the individual competencies of their components (which can perform tasks in their own space without any inkling of the large-scale goals to which they contribute) (**c**). This integration enables collectives to survey larger spatial areas and transmit information about possible solutions from one individual to another and likewise expands the capacity of the individual to incorporate information from the past and anticipated future due to the greater computational capacity and broader spatio-temporal perceptual/actuation horizons.

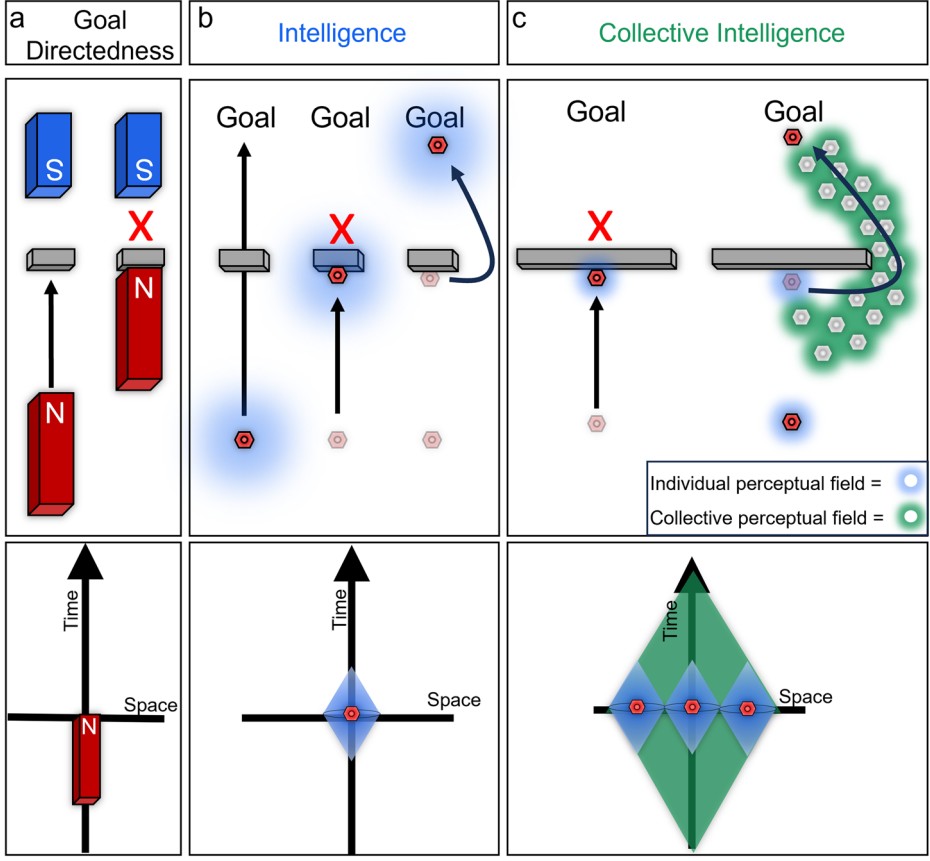

traits). We extend James' definition of intelligence to collectives by considering the perceptual field of an agent: the area in space and time that the agent can survey to find alternative paths to a goal (Fig. 2a, b). As the size of a collective increases its perceptual field increases, improving its ability to find variant paths. (Fig. 2c). It should be noted that while we here focus on animal development, there are also fascinating data of this kind in plants[53,56–61].

The most familiar examples of collective/swarm intelligence are beehives, ant and termite colonies[62,63], and flocks of birds and fish[64–66]. However, it is important to keep in mind that there is no sharp distinction between these collective minds[46] and putative centralized ones like those possessed by complex animals and ourselves[67] – instead, the biosphere offers a spectrum

of architectures including familiar solid brains where the neurons do not move much relative to each other (except in exceptional cases such as metamorphosis[68]) and so-called "liquid brains" – constructs in which the subunits can implement fluid interactions[63,69]. Fundamentally, typical brains are a collective of neurons, and provide an experience and functional unity of memories, goals, and preferences because of their interaction dynamics. Thus, one way to view cognitive science is as the study of the collective Intelligence of neurons *and other cell types*. Understanding how collectives ensure cooperation toward adaptive ends in diverse problem spaces is as much a part of understanding ourselves as of understanding ant colonies. Indeed it has been hypothesized that the remarkable ability of neurons to

unify toward a centralized self – the emergent agent that is the subject of memories, preferences, and goals which are not assignable to any of the individual components – is an evolutionary pivot of far earlier cell communication strategies that first solved problems in navigating another domain that requires information processing above the single cell level: anatomical morphospace[70]. By exploring possible scale-free dynamics in diverse systems, such as viewing the processes of morphogenesis as a kind of behavior of cellular swarms in anatomical space, we may enrich both behavioral neuroscience and developmental/regenerative biology by an influx of new ways of looking at the data[28,71,72].

Crucially, despite the clear parallels to the neuroscience of cognition, we here do not make any claims about first-person experience of unconventional collectives[73–76], nor are we saying that the phenomena we describe are of the same *degree* as familiar human-level cognitive capacities. Instead, we aim to take developmental biology and evolution seriously, and investigate the plesiomorphic, necessarily much more minimal, versions of decision-making and other proto-cognitive functions in multicellular contexts such as regeneration, development, and cancer. We view this as an important step to unify the recent progress in studies of single cell perception/action loops[77–80] with the work on active inference and perceptual control theory currently being developed in neuroscience, robotics, and artificial life[81–87]. Specifically, it is essential to move beyond low-level models of information processing, memory, and anticipation in chemical pathways[88–90] or in single cells[91–94], to understanding the higher-level perceptual landscape of multicellular collectives[95,96]. This knowledge is essential to improve our ability to explain, control, and re-engineer complex morphological and functional outcomes that today are still outside of our reach[25,26]. Here, we review interesting examples of the early, simple precursors to the processes that could underlie complex cognitive architectures, and we explore the hypothesis that the porting of tools across disciplines (dissolving artificial barriers between fields) may facilitate further research.

### Metazoan somatic cells make collective decisions
Development: transition to multicellularity. A most fundamental example of collectivity is observed during embryogenesis (Fig. 3a). When we observe a blastoderm, we call it an embryo. What precisely are we counting when we say it is 1 embryo? One answer is that the system consists of subunits all cooperating toward a specific path in morphospace: the cells are committed to making a specific functional anatomy corresponding to 1 individual. In effect, we are noticing alignment – both physically, in the sense of planar polarity of cell orientation in a collective[97–99], and functionally, as is seen in regulative development: if perturbed, the processes of anatomical homeostasis[28,100] will attempt to correct and compensate, toward a specific outcome considered normal, across a range of circumstances that includes, but is not limited to, standard development[101,102].

Indeed, the question of how many individuals are present in an embryonic blastoderm is not fixed at 1 by the genetics, because temporary introduction of breaks in the blastoderm (leading to informational isolation of islands of cell masses) results in twins, triplets, etc[103]. (Fig. 3b–d), showing that the blastoderm is a dynamical excitable medium in which multiple coherent embryos can self-organize. The same is true of organogenesis, such as when an induced large eye-field fragments into a number of individual eyes instead of one large eye (Fig. 3e). Thus, the physiological process that leads to the emergence of integrated collectives, which scientists and conspecifics recognize as discrete individuals is fundamentally dependent on the geometry of interactions (and signaling barriers) present during the early establishment of individuality and the setting of borders between Self and outside world (since every cell is some other cell's adjacent neighbor).

Cell migration: the group does not go where each of the parts wants to go. One major question about the origin of higher-level individuals from active components (cells, which are themselves not passive agents[9,21]), is how the behaviors of the higher levels depend on those of the lower-level components. The most obvious scaling mode is linear: the collective does what its individual cells are doing. But a more interesting aspect is that the collective often displays new behaviors or preferences. One example of this in the same space concerns cell migration. In an electric field, keratocytes migrate to the cathode, but *fragments of keratocytes* migrate to the anode[104]. Remarkably, individual fragments have

---

## Box 1 | on proto-cognitive terminology applied to unconventional systems

The use of the word Intelligence and other cognitive terms applied outside of its familiar context of brainy animals immediately raises questions: might not these terms be misused? Are not morphogenetic systems simply following the rules of chemistry – why anthropomorphize them? This is a crucial question. First, in the modern age, we must accept that *all* cognitive systems – ourselves included – exhibit chemistry, not magic, when one drills down to examine the lower levels. Thus, there simply is no special human category which one can correctly anthropomorphize as somehow being beyond the laws of physics at its base. We argue that this word is an anachronism and needs to be retired in favor of an empirically-grounded view, updated with the latest findings in causal information theory[205–210,233], in which it is perfectly possible (in fact, unavoidable) for a system to both, be subject to chemistry, and also to possess additional levels of description and control whose recognition affords novel benefits. We offer two points in clarifying our use of this terminology (developed in detail in[3]).

An uncontroversial aspect of our view is that claims of intelligence (and other cognitive terms), like all others, must be based on rigorous experiment, supported or ruled out by the degree of objective benefit that a given framing affords in terms of a) prediction and control, and b) future discovery (and new research programs) it suggests. The latter is most significant, because almost any paradigm can be rescued by enough epicycles; indeed, after one has discovered a new effect or reached a new capability, it is easy to drill down to the chemistry and – looking backwards – claim that there is no intelligence here because it mechanically

follows the laws of physics. The same is true for any act of a complex human brain-body system – if one insists on a view from the level of particles, it will always be there. The key question is: does that level of perspective provide the most interesting platform from which to make the next discovery or develop the most effective control policy. The emphasis should be on novel capabilities, and new research programs facilitated (or suppressed) by a given perspective. Thus, we propose that attempts to mine the rich toolbox of behavioral science to understand and exploit capabilities of morphogenetic systems will continue to pay off in many (but no doubt, not all) cases. We have fleshed out the prior gains facilitated by this view, and the promises for regenerative medicine, elsewhere[16,17].

The less conventional, and sometimes uncomfortable, aspect of our position is that the empirical utility of framings needs to be applied fearlessly, and followed wherever it may lead: its empirical consequences must be taken seriously even when they contradict long-cherished a priori commitments to how non-intelligent a given system must be. In other words, if a specific framing, which uses tools normally reserved for brains, results in fruitful new research programs on bacterial biofilms[35,36,38], plant roots[56,57,59–61,234,235], the training of gene-regulatory networks[225,226,236,237], or developmental/regenerative biology[117], then the scientific approach requires that we consider those systems to be bona fide subjects of that corner of the natural world that is supposed to be described by the behavioral science of a spectrum of minds.

---

**Fig. 3 | Embryos as emergent, physiologically-defined collective entities. a** Embryonic blastoderms are considered "1 embryo" even though they are made of thousands of cells because the cells are aligned – both physically, in the sense of planar polarity, and behaviorally, in the sense that they will all cooperate toward one attractor state in anatomical morphospace. However, if the blastoderm is temporarily scratched (a'), each island will, because it doesn't feel the presence of the others until the scratches heal, form its own embryo[103]. This results in conjoined twins, triplets, etc. such as these two duck twins shown in **b**. Indeed (**c**), this fascinating process demonstrates that 1) the number of embryos emerging from the excitable medium of an embryonic blastoderm is not genetically fixed but determined in real-time by physiological processes, and 2) involves a fundamental process of collective autopoiesis in which each high-level individual (e.g., an embryo) needs to determine the borders between it and its outside world. **d** Such collective decision-making, which regulates the behavior of the sub-units, and its failure modes, is starkly revealed as conjoined twins are known to often exhibit laterality defects (such as heterotaxy) as cells located between two self-organizing embryo collectives may not always decide correctly whether they are the left side of one twin or the right side of the other[229]. **e** The same task is solved by individual organ primordia, for example when ectopic eyes are induced by ion channel misexpression[150], forming several distinct eyes of normal size instead of one giant eye. Panels **a** and **c** were made by Jeremy Guay of Peregrine Creative. Panels **b** and **e** produced by co-author M.L. (Tabin lab) and Sherry Aw (Levin lab). Panel **d** is used with permission from[229].

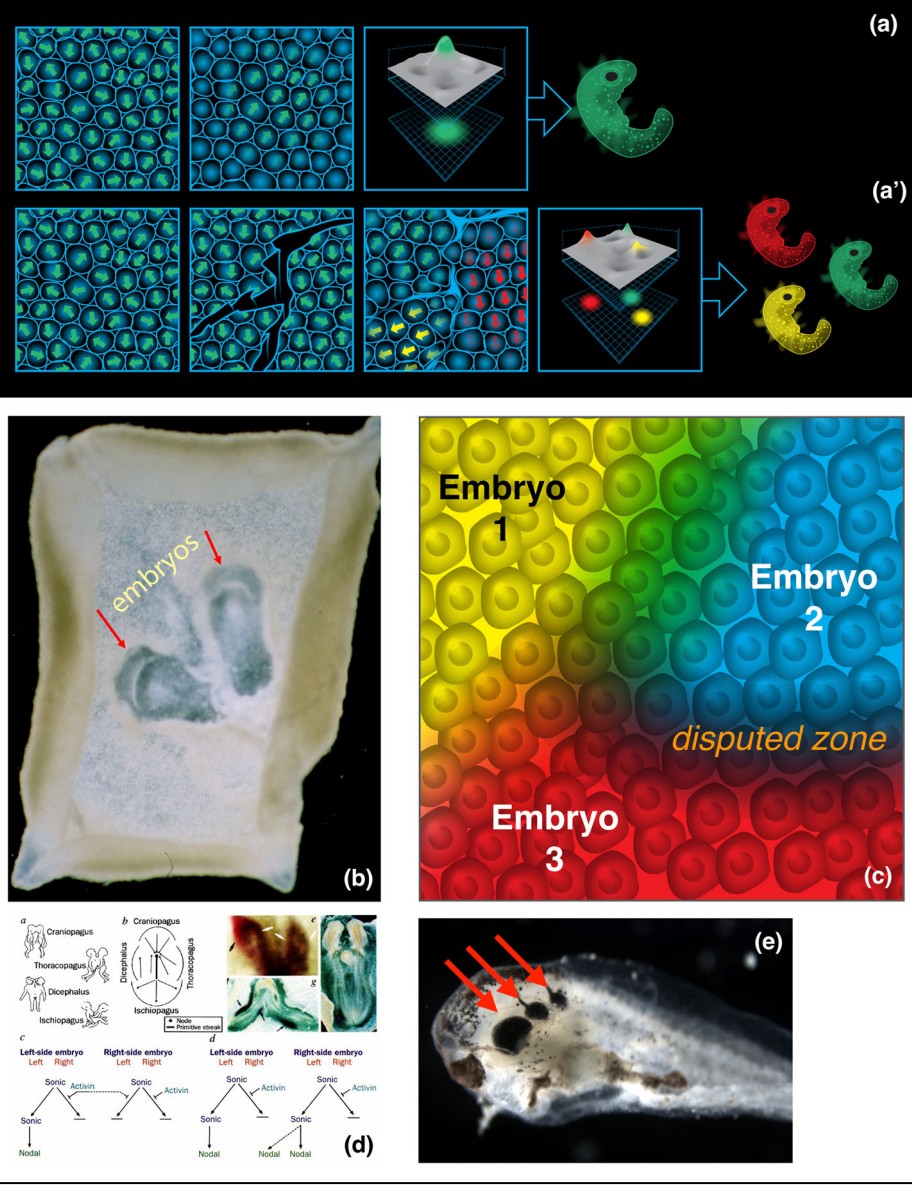

the opposite direction of taxis to that of a collective of those fragments (an intact cell). The behaviors of a collective can, even in relatively minimal systems, be a complex and hard-to-predict function of the tendencies of the components. This is a microcosm of the larger issue of competition between wholes and their parts[6,105], and of the more general feature of multiscale organization in which collective agents bend the energy landscape for their components to exploit their mechanisms towards distinct ends.

**From normal melanocytes to a melanoma phenotype: a collective decision.** One failure mode of collective behavior in vivo is cancer[106,107]. When cells become isolated from the information structure of the tissue, they revert back to an ancient, unicellular transcriptional[108] and behavioral phenotype[109,110]. Exciting work focusing on the biochemical nature of the microenvironment has shown the ability of non-cell-autonomous cues to normalize cancer[111–116]. However, more recent work has focused on bioelectric cues that normally orchestrate multicellular anatomical outcomes[117], and the consequences of their disruption (Fig. 4).

In the tadpole model, it was shown that normal melanocytes could be driven into a melanoma-like converted phenotype: they over-proliferated, migrated inappropriately to regions normally devoid of melanocytes,

invaded the blood vessels and brain, and changed shape into a highly arborized, invasive form[118]. This could be achieved in the absence of classical carcinogens, oncogenes, or DNA damage, by brief exposure to chemical (chloride ion channel activator drug) or molecular-genetic (GlyCl mutant) targeting of the bioelectric state of a specific cell population: instructor cells which normally keep the melanocytes in their healthy state via serotonergic signaling[118]. Indeed, targeting only a handful of instructor cells in a region away from the source of melanocyte populations (as confirmed by lineage label) was sufficient to turn the whole tadpole into a hyperpigmented phenotype strongly resembling metastatic melanoma: all of the melanocytes converted, even the ones not close to the GlyCl-activated cells.

The most remarkable thing was that this phenotype is an all-or-none phenomenon. Using different reagents could induce different incidences of hyperpigmentation (conversion) in a *cohort* of animals, but this was a population-level phenotype: for example, 70% of the animals could be converted, but any given animal was either fully converted or fully normal (Fig. 4a, b). A computational model of the known signaling steps was designed and parametrized (Fig. 4c) to reproduce this all-or-none behavior and fit the experimentally-observed incidence percentages across different perturbations[119]. The model illustrated how cells navigate biochemical state space (Fig. 4c') and face specific decision-points at regions of that landscape.

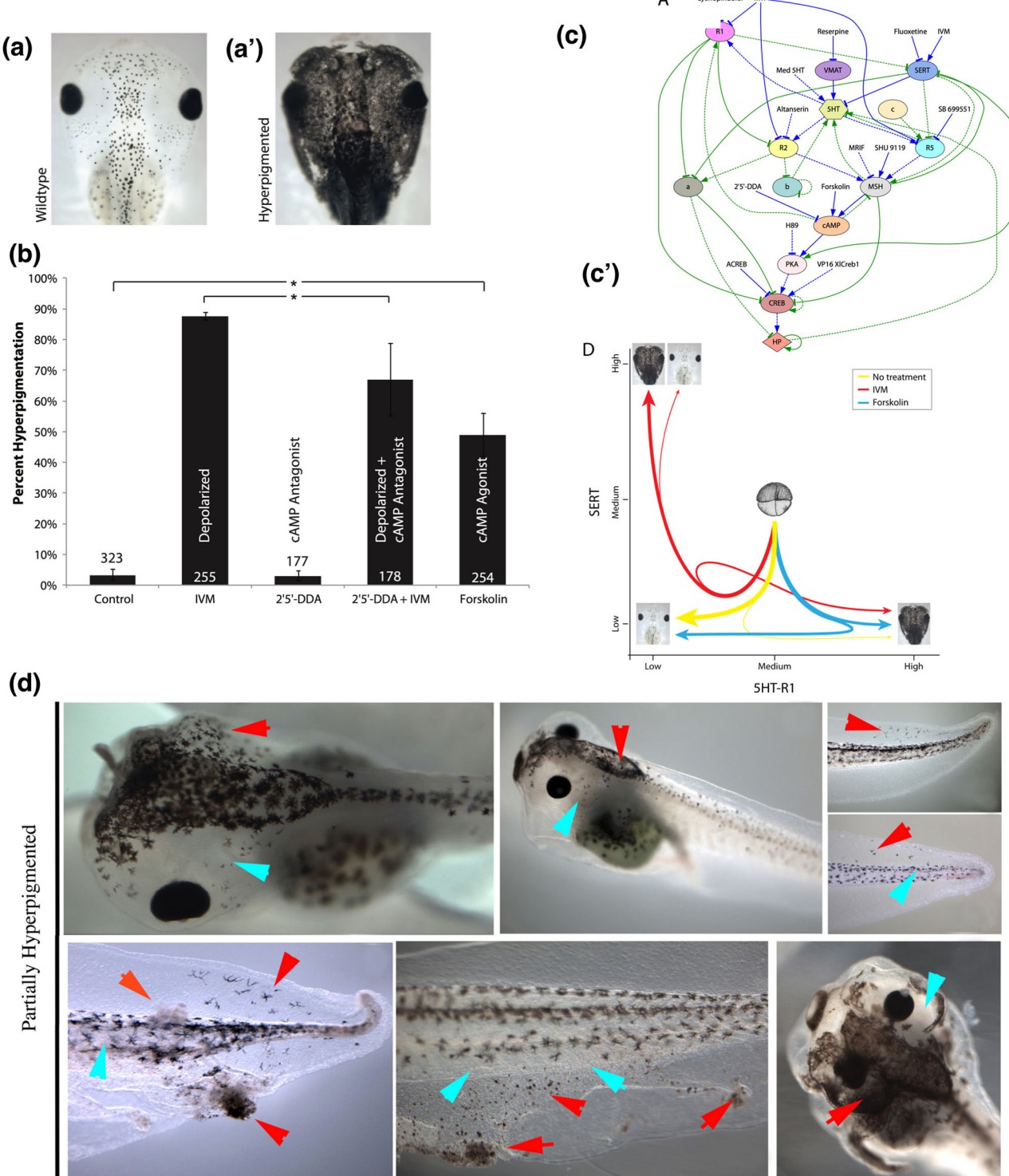

**Fig. 4 | Collective decision-making of neoplastic transformation.** Tadpoles of the frog Xenopus laevis (dorsal view of the head) show small numbers of round melanocytes (**a**) in normal development. However, when the signals from instructor cells[118] are interrupted by changing their resting potential, the melanocytes convert to a melanoma-like form: they hyperproliferate, and migrate to fill many regions of the embryo in a phenotype that recapitulates melanoma metastasis (**a'**). This process can be stimulated at different rates in a population by different reagents (**b**). Interestingly, while only some percentage of animals convert, the decision is made in a coordinated fashion by every tissue in the animal: individuals are either entirely converted (every melanocyte undergoes the shape and behavioral changes) or are entirely unaffected. This stochastic decision resembles a biased coin toss, but all the tissues of a given animal are linked to the same coin. A model of the pathway (**c**) was parametrized by a machine learning algorithm which was able to reconstruct the state space (c') revealing the mechanism of this stochastic decision[119,120]. This model was then used to predict a novel intervention that would *break the concordance* among cells – disrupt specifically the collective decision-making. For the first time, partially-pigmented animals were produced (**d**), showing how AI-discovered pathways fitting biological data can help explain collective decisions made by large numbers of cells in vivo. Panels a-c' taken with permission from Ref. [119]. Panel d taken with permission from Ref. [120].

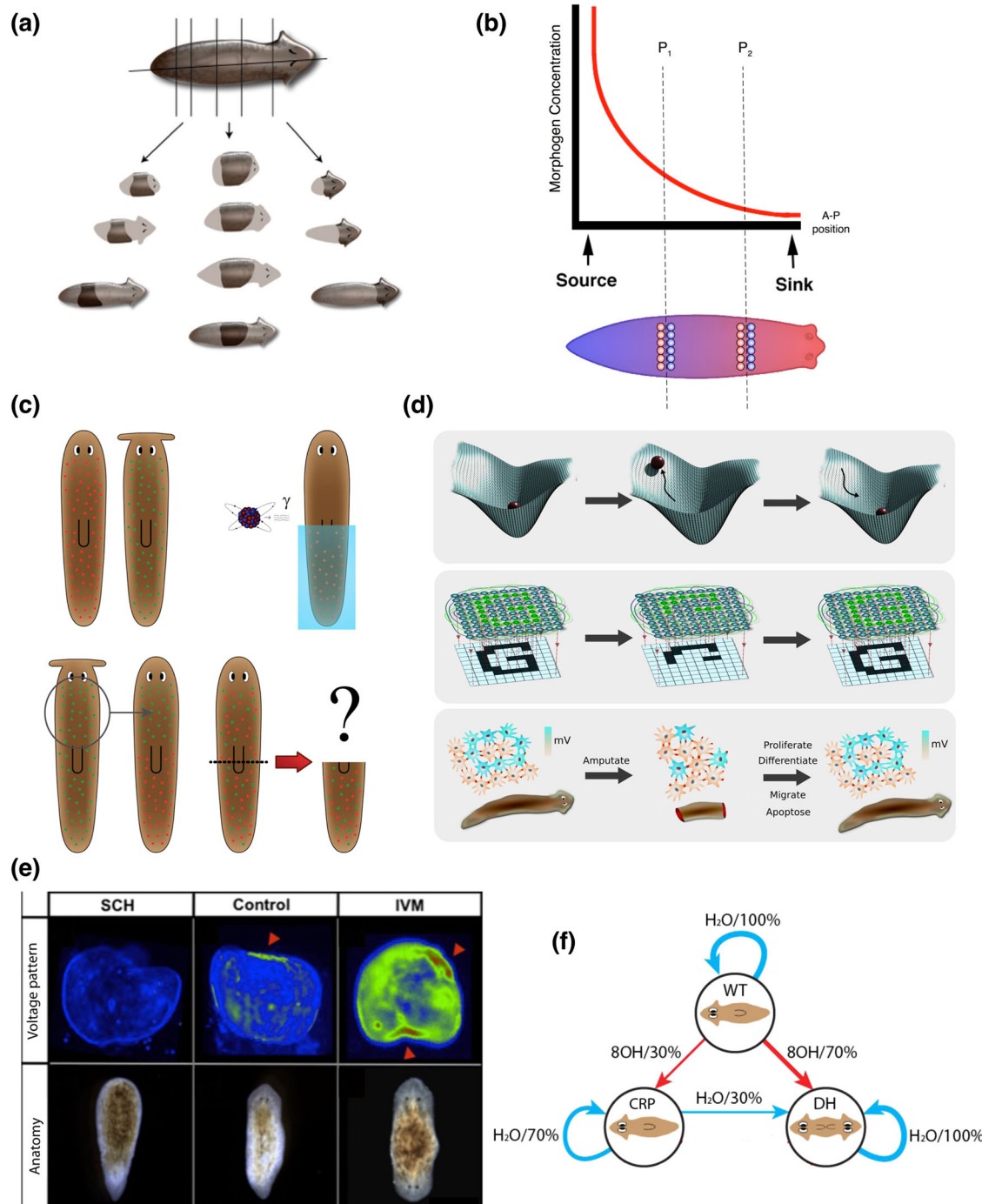

The benefit of such a model of collective decision-making is that it can be used to infer interventions. Specifically, the model was used to predict an intervention that would *break the concordance* of melanocytes within single animals. It suggested two drugs and a dominant negative construct – an experiment that had never been done before - which were then experimentally confirmed to produce the first partially-converted animals (Fig. 4d) seen in almost a decade of experiments in this system[120].

**Left/right, head/tail: random, but collective anatomical decisions.** In addition to the 3D and transcriptional/physiological spaces discussed above, one of the most interesting aspects of collective intelligence is the navigation of anatomical morphospace. Cell groups need to make specific decisions about which organ will be built and what shape they must make.

This is a fundamentally different problem than identifying gene regulatory networks and differentiation signals. For example (Fig. 5), planarian cells can rebuild a complete worm from any kind of cut or fragment (Fig. 5a). Typical treatments of this problem focus on a fragment within a morphogen gradient, that offers distinct concentrations of an instructive chemical signal that can confer head/tail fate decisions locally to each wound (Fig. 5b). However, the more interesting and fundamental issue is seen when considering just one cut: the cells on either side of the cut will create a head and tail respectively, but they were adjacent neighbors before the cut and located at the same positional information value. In other words, it is actually impossible for an anatomical decision like this to be made locally – the cells of the wound must coordinate with the remaining fragment to get information about where

**Fig. 5 | Regeneration: collective behavior in anatomical morphospace. a** Planaria regenerate a complete worm from even small fragments, which requires large numbers of cells to cooperate to complete a specific path through morphospace, eventually building a correct complex anatomy to very tight tolerances. **b** Morphogen models are often used to explain the axial polarity of a middle fragment (2 cuts), but if one focuses on a single cut (vertical line marked as P1), it is clear that adjacent cells will have radically different anatomical fates (head vs. tail) despite their identical positional information. The cells must communicate with the rest of the fragment to determine what other structures exist, which way the wound is facing, etc. and then decide which path in morphospace the collective will follow. **c** If half of the neoblasts of one species of planaria (with a round head) are killed by radiation and replaced with transplanted neoblasts from another species (with a flat head), what shape head would regenerate (and would it ever reach the stop criterion and cease remodeling)? This thought experiment reveals clearly why the science of collective intelligence is a critical complement to molecular genetics: while much single cell-level information is available about the pathways controlling stem cell differentiation, the field has not a single model able to make a prediction for this scenario, because we still lack the conceptual tools and data to understand how collectives of cells make unified decisions. **d** The tools of dynamical systems theory (top row) and connectionist neuroscience/AI (middle row) are poised to help provide formalisms for understanding how networks of cells can store pattern information and recover it from partial inputs, such as occurs in planarian regeneration (bottom row). **e** In planaria, one of the modalities that binds individual cells into morphogenetic collectives is bioelectricity[117,132]: fragments possess a difference in resting potential that determines the number and location of heads. Tracking these patterns using voltage-sensitive fluorescent dyes in functional experiments reveals some aspects of the rules by which the collective makes decisions[230]. In animals

treated with the proton pump inhibitor SCH28080 (SCH column), the bioelectric pattern lacks the depolarization that cells interpret as the make-a-head signal and headless animals result (bottom left). Control fragments have the depolarized (green) signal at one end (middle column) leading to normal 1-headed animals (bottom center). Animals treated with the chloride drug opener Ivermectin (IVM column) exhibit two regions of depolarization resulting in 2-headed animals (bottom right). Interestingly, the green voltage that produces heads in control animals is seen in the middle of the IVM-treated worms, and does not induce heads: only the very depolarized regions (red) become heads. This indicates that the collective is not measuring absolute resting potential values, but (in keeping with the distributed nature of the circuit and the animal-wide signaling) is adopting anatomical organ-level fates driven by the *relative difference* of regions (i.e., the *most-depolarized* region is where heads form). **f** In addition to 0-, 1-, and 2-head worms (which are stable lines that continue to regenerate as 2-headed[231]), there is another form called Cryptic Worms, also produced by bioelectrical disruption[130]. These worms show a stochastic phenotype, in which a worm (or indeed, independent pieces of a single worm) will form 2-head or 1-head worms at a 70–30 ratio; the transition diagram with probabilities is shown in panel **f**; WT = wild-type (1 head); 8OH = octanol which causes the cryptic phenotype; H2O – water (control condition or cutting). The percentages indicate the frequency of each transition. Crucially, while the head/tail decision is stochastic, all of the cells agree on the same outcome (what is never seen is a planarian in which some of the cells in a given region are making a head and others a tail) – the whole region makes the random decision as a single whole. The mechanism is not known but likely involves gap-junctional communication of the voltage signals[231] and can be modeled as a kind of perceptual bistability[131]. Panels **a** and **d** made by Jeremy Guay of Peregrine Creative and Alexis Pietak respectively. Panels **c**, **e**, **f** used with permission from Refs. [121,130,230] respectively.

---

they are located, which way they are facing, and what other structures exist[121,122], in order to make adaptive decisions about large-scale growth and form that enable regeneration of normal worms.

More generally, numerous excellent papers have studied planarian neoblasts and their control networks, as well as the gradients of morphogens that pattern the anterior-posterior, dorso-ventral, and medio-lateral axes[123–127]. Despite these advances, there is very little understanding of how cells build specific head shapes or how they know when to stop mitosis and morphogenesis when the correct head shape has been achieved. Specifically, for example, no existing model makes a prediction on what will happen if 50% of the neoblasts of a given planarian are replaced with those of a different species and the head is cut off (Fig. 5c). Whether the head will be of the right shape for one of the two species (dominant), or an in-between hybrid form, or in fact continuously cycle between shapes (as each set of neoblasts works to remodel toward the shape they normally make with great fidelity), cannot yet be derived from the properties of single cell regulatory pathways – it is a collective decision about navigating the space of possible head shapes[128,129] (Fig. 5d).

Indeed, modification of cell:cell communication during regeneration can cause genetically-normal fragments to produce heads appropriate to other species of planaria[128,129] – visiting attractors in morphospace normally reserved for other genetic lineages. More specifically, several perturbations targeting the bioelectric control circuit (Fig. 5e) have shown randomization of outcome: such Cryptic planaria are destabilized, and fragments (even from the same parent worm) will form 1-head and 2-head forms at a set frequency of ~1:2[130] (Fig. 5f). This phenomenon highlights *collective* decision-making because this randomization is at the level of the population: each individual animal has clear heads and tails, not tissue speckled with cells of different identity. In other words, the randomization of bioelectric state[131,132] and the downstream morphogen gradients is interpreted with respect to anterior-posterior organ identity by collectives, not by individual cells.

The left-right axis in vertebrates shows a similar phenomenon (Fig. 6). Consistent asymmetries across the midline first show up in the chick embryo around the primitive streak and Hensen's node[133]. A number of treatments, including targeting of the bioelectric[134–138] or downstream biochemical[139–142] pathways, result in randomization of molecular and anatomical consequences of symmetry breakage and orientation[143–148]. The animals display, in addition to the normal L:R identity, double-right or

double-left (isomerism), or reversed (situs inversus) patterns of lateral identity markers, followed by heterotaxy of the heart and viscera. Remarkably, while many of these treatments randomize outcomes, the randomization is once again above the level of the individual: any given embryo has a consistent identity on the L and R side, and all of the cells agree. In all of the many studies on perturbation of the LR pathway, we are aware of only one that actually breaks the concordance: disruption of the planar polarity pathway by down-regulating VANGL signaling[149] leads to a speckling on both sides of the midline, consistent with *individual cells* within a single lateral domain disagreeing on whether they should have L or R identity (Fig. 6a–d).

## Eye or skin: competition in DNKir6 injected animals, and size control

A number of collective decisions are mediated by bioelectric signaling, which coordinates cells in the body as a likely precursor to its role in coordinating neurons in the brain toward the emergence of a coherent, problem-solving Self[27,70]. One example of this at the organ level concerns the induction of whole ectopic eyes in the frog embryo by misexpression of ion channels[150] whose activity sets up a voltage gradient similar to that of the eye spot which normally determines their location in the head[151]. As with previous examples, this is a signal to the collective, setting organ-level identity, not micromanaging the differentiation of the many cells which need to be produced and placed with exquisite precision to make a normal vertebrate eye. Interestingly, this signaling has another built-in competency: recruitment. If very few cells are injected with the channel (attached), they will often recruit their neighbors (Fig. 6e, f) to help them complete the task. This is a kind of secondary instruction, where we instruct a group of cells to make an eye, and they recruit the others (which were never directly manipulated), including all of the necessary downstream morphogenetic steps. This recruitment of individuals to accomplish a high-level goal is seen in other collective systems like ant colonies[152,153], which often call in helpers when a task is large. The ability to recruit participants to complete tasks may be a central competency of collective intelligence that works across scales, from cells to swarms of entire organisms[7].

## The neural crest acts as an intelligently migrating collective

The neural crest is a cell population that arises between the neural plate and the non-neural ectoderm before migrating throughout the body to produce

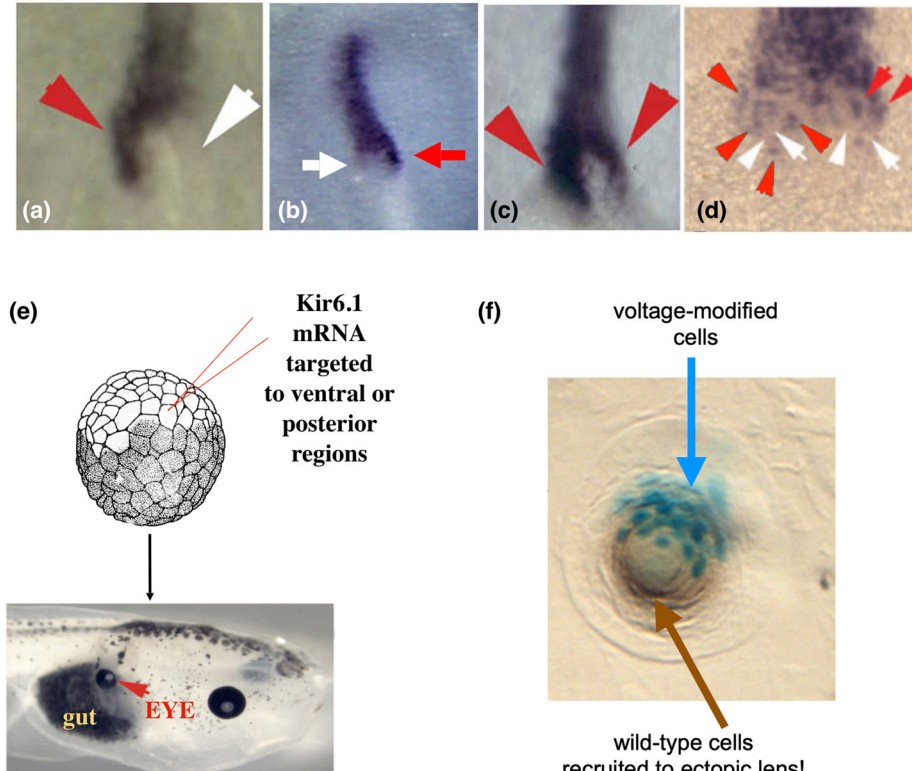

**Fig. 6 | Cellular collectives coordinate decisions from body axes to whole organs.** The expression of Sonic hedgehog in the Hensen's node of early chick embryos is a marker (purple color) of left-right asymmetry: in normal embryos, it is only expressed on the left side (a, red arrowhead). By perturbing upstream events, such as the voltage gradient that determines lateral identity[138], it is possible to induce a percentage of right-sided (**b**) or bilateral (**c**) expression. However, in all these cases, all of the cells on one side of the node agree on their identity, even if the identity is stochastically determined within a cohort of embryos. There is one known perturbation – disruption of planar polarity[149] – that breaks the concordance and results in a speckled appearance (**d**) in which individual cells within a lateral compartment disagree about which side they are on. **e** Ectopic eyes can be produced in frog embryos, even far from the anterior neurectoderm such as on the gut (red arrowhead), by microinjection of potassium channel mRNAs that induce a specific state that is interpreted by cells as an eye induction signal[150]. The information content of

this inducer is very low – as in the endogenous eye spot in the head[151], there is a simple voltage pattern that cannot encode all of the nuances of a complex vertebrate eye structure. This simple spatial pattern is read out by the cellular collective to make an organ-level decision. Interestingly, if very few cells are injected (**f**), the blue cells are lineage-labeled to indicate the presence of the ectopic ion channel), normal-sized lenses can be produced because the voltage-modified cells recruit their wild-type neighbors to participate in the organogenesis. However, this does not always work because the surrounding cells, in a cancer-suppression mechanism, are meanwhile exerting influence to shut down the ectopic eye induction, and sometimes no eye tissue appears at all. This is an example of collective decisions competing to determine the path through morphospace that will be taken and once a choice is made, all of the cells fall into line. Panels **a**, **c**, **d**, **e**, **f** taken with permission from Ref. [3,149,150,232] respectively.

a constellation of cell types including head mesenchyme, peripheral nervous system and melanocytes[154]. Neural crest cells (NCCs) must successfully traverse the complex and rapidly changing embryonic body before identifying their target location and integrating with nearby tissue. The energy landscape that they traverse is more complex than it initially appears, however. While each cell navigates a fairly simple dorsal-ventral cartesian space towards a goal destination, the neural crest collective is navigating morphospace to create properly spaced, symmetrical facial structures. When the cell-level navigation of cartesian space is put in opposition to the collective-level navigation of morphospace, the collective supersedes the behavior of the individual to achieve the organism-level morphogenetic target of forming a functional, symmetrical face as we describe in the examples below.

**Neural crest migration is intelligent.** Grafting and ablation experiments underscore the collective ability of the neural crest to accomplish its morphogenetic goals despite some novel circumstances. Axolotls regulate the number of cells, compensating for too few or too many[155]. Neural crest cells (and in some cases neural tube cells[156,157]) regulatively adapt their migratory behavior to compensate for the loss of NCCs in nearby or contralateral branchial arches[158,159]. This re-routed migration suggests that individual cells can leverage the perceptual field of the

neural crest cell collective to determine the movements that they should take to contribute to proper system level morphogenesis (Fig. 7). In an especially striking example of the NCCs' intelligent capability to achieve their ontogenic goals in challenging environments, mouse NCCs grafted into chicken embryos will successfully navigate the forming embryonic face and form teeth[160].

**Neural crest intelligence is collective.** Individual cells transposed from one anterior-posterior axial domain to another will change their gene expression to match their neighbors[159,161,162]. In contrast, groups of cells transposed along the anterior-posterior axis maintain their original gene expression, thus resisting the inductive effects of the surrounding tissue[159,161,162]. Within the context of our perceptual field model, the increased positional memory and resistance to neighbor effects suggests that cell collectives have an expanded perceptual cone in the posterior time (history) dimension (Fig. 7). While individual cells rapidly lose their memory of past inductive cues, collectives are better able to maintain a consistent identity in a noisy developmental environment[159,161,162]. As with the re-routing post arch ablation example above, this example suggests that the cell-level behavior is subordinate to the collective-level behavior. When collective-level behaviors are put in opposition by grafting of collectives these is no clear hierarchy, and original fate is maintained.

## Cells intelligently segment the vertebrate body axis via the segmentation clock

Another well-studied example of collective intelligence is the vertebrate segmentation clock comprising coordinated oscillations of Notch pathway target genes to establish segmental boundaries of the early vertebrate embryo[163]. The segmentation clock exhibits functional robustness to interventions, consistent with James' definition of intelligence, because of the ability to take different paths through morphospace to correctly partition tissue into uniform, correctly sized segments. Clever experiments radically altering the geometry of the tissue test its intelligence by forcing it to explore a variant morphospatial landscape.

**Segmentation is intelligent.** In whole embryos, coordinated oscillations will re-emerge following chemically induced disruption and resume producing properly spaced segments[164], though the complexity of the developing organism make it challenging to determine if this re-emergence is intrinsic to the segmenting tissue or imposed upon it by other tissues. Recent work with paraxial mesoderm explants[165] further emphasizes the remarkable intelligence of this system. In the context of James' framework for intelligence, the goal states are 1) coordinated oscillatory gene expression and 2) morphological segmentation, and the obstacle is the severe geometric transformation from 3D tube to 2D sheet. In the 2D geometry, oscillatory gene expression waves manifest as outwardly propagating rings that successfully effect segmentation of the outer edge of the explant. While these segments manifest as serially repeated spheres in the 3D in vivo environment, 2D cultures form segments circumscribing the explant's circumference (Fig. 8a). Segmentation can be re-capitulated from embryonic stem cells in culture by production of trunk-like organoids termed gastruloids[166,167], which arrive at a segmented target morphology despite a very different ontogenic history than normal trunk cells.

**Segmentation intelligence is collective.** The ability of the segmentation clock to intelligently navigate its morphospatial landscape has also been tested by forcing it into a state that it would never normally adopt by grafting out of phase cells into oscillating tissue. The segmentation clock functions to coordinate a collection of cells to organize into a large supercellular structure. Consistent with this function, collectivity is necessary for the segmentation clock to function. When wild type cells are grafted into mutant non-cycling fish, they express the normally oscillatory gene *her1*, but it does not cycle[168]. Similarly, pre-somitic mesoderm cells do not oscillate when cultured independently, but will resume oscillating when cultured collectively[169]. This loss of oscillation can be partially rescued by addition of external FGF, potentially mimicking the effects of high cell density[169] and implicating collectivity in stem cell maintenance. Most directly, cells hetero-grafted from tissue in one phase of the clock into a group of cells in a different phase will synchronize to the phase of their lateral neighbors[170], (Fig. 8b). In the context of our perceptual field model, the grafted cells benefit from the expanded memory and predictive power of their neighbors to determine their correct position in the clock (Fig. 8b). These cells then adjust their intrinsic oscillatory dynamics to entrain to their neighbors, thus completing their task despite an internal configuration with novel hardware components which do not have an evolutionary history of living together in a single organism.

## Intelligence in bacterial communities

Though bacteria are unicellular, they often form into large biofilms that exhibit fascinating physiological and morphological collective properties[171–174]. Interestingly, much as bioelectric networks are used in metazoan systems to bind individual cells together to large-scale morphogenetic projects, bacteria likewise exploit electrical signaling across space and time to coordinate[175,176]. Cells within the biofilm (and even between biofilms) exhibit bioelectrically-coordinated oscillatory growth patterns that favor the health of the collective at the expense of their own individual fitness[177,178], and bioelectric signals coordinate metabolism among distant

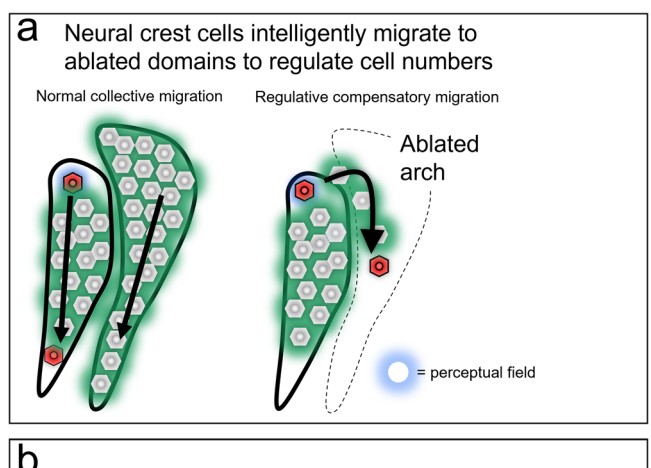

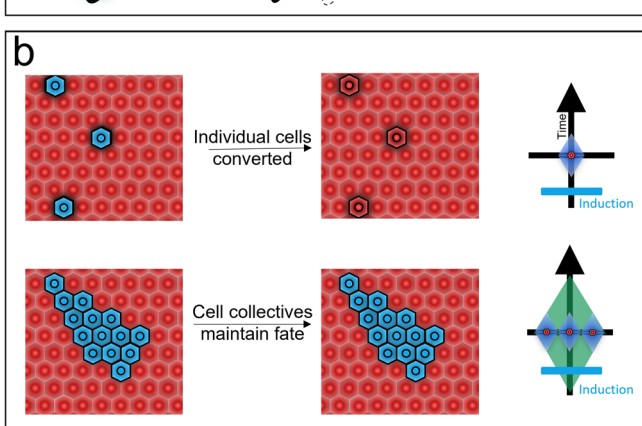

**Fig. 7 | Neural crest cell collectives intelligently alter the behavior of their component cells to achieve target morphology.** Regulative compensatory migration in migrating neural crest cells demonstrates collective intelligence by placing the goal of the individual and the goal of the collective in opposition (**a**). During normal development neural crest cells integrate signaling cues to migrate dorsal to ventral along the forming embryonic body. When an arch is ablated, cells will move anteriorly or posteriorly instead of dorsoventrally to fill this missing arch thus achieving the correct target morphology of the collective behavior through movement patterns contrary to the individual cell's normal optimal path[158,159]. Applying our perceptual field framework, the expanded collective perceptual field contains a more attractive path to achieving the cell's goal than its individual perceptual field does, and thus is undergoes compensatory rather than normal migration. Inter-domain grafting experiments demonstrate the expanded time-domain perceptual field of a collective intelligence (**b**). When grafted from one Hox domain to another, individual rhombomere or neural crest cells will lose the memory of their original Hox gene inductive event and adopt the expression pattern of their neighbors[161,162]. Similarly grafted collectives, in contrast, maintain their previously induced state suggesting increased memory due to their collective intelligence.

cells within the biofilm[38]. These bioelectric signals help recruit new bacteria to the biofilm, even across species[179], and can be optogenetically controlled to evoke long-lasting changes on bacterial behavior – a collective memory[35].

Exciting recent work has identified a mechanism similar to the vertebrate segmentation clock in bacterial biofilms responding to nitrogen stress mediated by a negative feedback loop[180]. The similarities between this system and the vertebrate segmentation clock point to further roles for this phenomenon in collective intelligence, and the manifestation of similar molecular logic circuits in distant clades suggests that such collective intelligence is a much more widespread phenomenon than is currently appreciated. Furthermore, the parameter space is neither cartesian space nor morphospace as in our previous examples, but physiological space. The bacterial cells intelligently adjust their individual physiologies to achieve an optimal collective physiology. The capacity of such simple organisms to collectively navigate physiology space using paradigms recapitulated in multicellular organisms highlights the deep cruciality of such navigation to

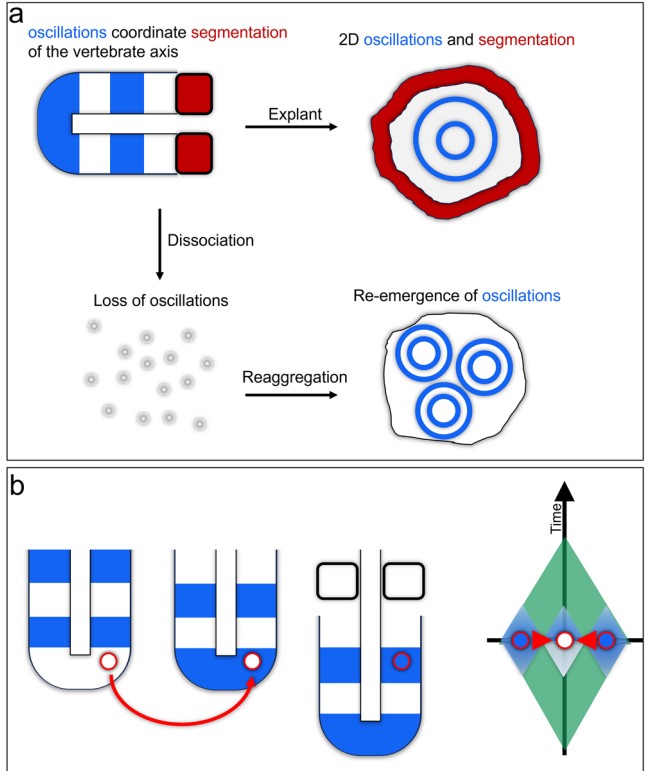

**Fig. 8 | Collectives of vertebrate paraxial mesoderm demonstrate intelligence in their ability to achieve their goal of segmentation despite being removed from developing animals and placed in a 2-dimensional culture system**[169]. When the tissue is dissociated, cells quickly stop oscillating, though coordinated oscillations re-emerge in cells that are dissociated and re-aggregated, indicating that collectivity is essential to this intelligent behavior[169] (**a**). Cells grafted between tissues in different phases of the segmental oscillator will synchronize with their neighbors in the host tissue. Within our perceptual field framework, this synchronization suggests that the system level memory (i.e., where they were in the oscillation) is conferred upon the grafted individual cells for whom the spatial and temporal cues are now mismatched (**b**).

the emergence of complex tissue and points to the necessity of understanding how such navigation occurs during animal development and pathology.

## Conclusion

Cell and developmental biology offer very rich fodder for the emerging field of diverse intelligence: discovering a vast spectrum of problem-solving capacities in novel substrates and at unconventional spatiotemporal scales. Because of life's multi-scale competency architecture, a fundamental aspect of intelligence is collective behavior: all intelligences appear to be made of parts, connected by mechanisms implementing policies that bind the competent components into a cooperative (and competitive[6]) computational medium that solves problems in new spaces and at higher scales. The harnessing of individual cell behaviors toward regulative morphogenesis (navigating anatomical morphospace), and system-level physiological robustness (traversing physiological space) are especially interesting examples. Indeed, it could be argued that a unique signature of Life is a causal architecture in which the problem-solving competency of the whole is greater than that of its parts). Evolution seems to be particularly good at finding ways to scale the cognitive light cone of cells[3–5,181] to achieve spectacular capabilities for gracefully and adaptively handling complexity, novelty, and noise at large scale.

A key aspect of collective intelligence of cell groups is binding subunits' activities to the same target morphology – a kind of discrete (e.g., head vs. tail) outcome whereas the components have states that range over many continuous quantities. In axial patterning (left-right,

anterior-posterior), collective decision-making enables large numbers of cells in a compartment to agree on an organ-scale anatomical fate despite stochastic influences upstream. And it is seen that a decision with respect to morphogenetic outcome, and harnessing cells to the same decision, are orthogonal functions with distinct mechanisms that can be experimentally dissociated.

Importantly, the definition of intelligence as the ability to reach the same endpoint despite internal or external changes emphasizes not only robustness (successful use of novel navigational policies to overcome perturbations) but also its failure modes. Numerous ways of targeting of its sensory, memory, decision-making, or other components can de-rail the performance of a collective intelligence, resulting in birth defects and malformations. This is quite consistent with the proposed symmetry between the behavioral and developmental domains, because computational neuroscience and cognitive science are replete with interesting ways to think about how cognitive systems make mistakes. The use of tools and concepts across fields has begun, including attempts to understand cancer as a dissociative identity disorder of the morphological collective intelligence[109], the use of serotonin reuptake inhibitors and hallucinogens to perturb non-neural development[182,183], the modeling of the unstable phenotypes in planarian regeneration as perceptual bistability[131], and the finding that some visual illusions that plague vertebrate nervous systems are recapitulated in collective intelligences such as ants[184,185]. We expect that many concepts from the behavioral sciences that explain failures of learning, recall, Bayesian updating of dynamic signaling models, attention, arousal, and perception will find application in explaining and controlling defects in navigation of anatomical space towards healthy, optimal outcomes.

Another hallmark of collective intelligence is the ability of the higher-level agent to make decisions based on extended patterns of information. For example, in the frog embryo brain, it is the spatial difference in voltage between regions that drives downstream gene expression, not the absolute value of any cells[186,187]. In other words, cells have to read whole cell fields and recognize specific patterns to determine what to do – the collectivity is seen in the input, as well as the output, of cell groups' behaviors.

Future work is essential to understand how higher-order entities (organisms, organs, tissues, etc.) distort the energy landscape for their subunits, benefitting from their competencies to navigate spaces of which the subunits are unaware. This underlies the harnessing of cellular signaling and computational abilities to regulative development and regeneration, which implement organ-level homeostatic loops that keep large-scale order against cellular defections (aging and cancer[106,188]) and injury[189]. Living matter is a kind of agential material with the ability to propagate information across scales – a phenomenon which has many implications for evolution[9], and for bioengineering[21].

Many tools are becoming available which increase insight and cross-fertilization of approaches across disciplines. Examples include optogenetic interrogation of single cell[78,190–193] and embryonic[194–199] dynamics, as well as the very elegant electrotactic 'SCHEEPDOG' system which is able to precisely steer collectives of keratinocytes using patterned dynamic electric fields[200] that distinguish between collective and individual cell behaviors. In addition to technologies, important additions are conceptual tools, such as the active inference framework[201–203] and tools of causal information theory[204–212], which will have many applications in the biological sciences.

Future work in this area will also continue to be enriched by advances in the collective intelligence of animal behavior[46,213] as well as in the field of swarm robotics[214–217]. Additional directions for investigation include: how conflict (competition) is used for coordination in collectives[6,218], and how propagation of shared stress[181,219,220] and the sharing of cellular memories via gap junctions[4,109] establish higher-order individuals.

One of the most exciting aspects of this emerging field is the way in which collective intelligence serves as a focal point for exploring the symmetries between developmental biology and neuroscience[26]. This ranges from the use of cognitive science formalisms to understand morphogenesis and its disorders[55,131,221] to the questions of how many human Selves can be sustained by the excitable medium of a human

https://doi.org/10.1038/s42003-024-06037-4 **Perspective**

brain[67] and the parallels to the multiple bodies that can emerge from a single embryonic blastoderm[103].

Many of the same mechanisms (e.g., electrophysiological networks) and control policies are re-used by evolution to bind neurons to collective behavior and animal navigation of 3D space and to bind pre-neural cells to move the body configuration in morphospace[5,70]. Turing was prescient in studying both intelligence and the chemical basis of self-organization[222,223], as the problem of self-organization in familiar neural-based intelligences may have much in common with the problem of self-organizing a non-neural collective intelligence of morphogenesis[224]. If true, a number of fields can look forward to exciting advances. Cancer, a kind of dissociative identity disorder of the somatic collective intelligence[109], limitations in regenerative ability, and many physiological disorders could all be advanced by techniques that exploit not just the low-level mechanisms, but also the higher-level decision-making of life[16,17]. Neuroscience can benefit from a glimpse into the evolutionary past of the brain's remarkable capabilities, while developmental biology and bioengineering can borrow the practical and conceptual tools of neuroscience which is likely to be about much more basic principles than the function of classical neurons. Understanding how evolution works in an agential, multiscale material (where it can take advantage of cross-level computation) will nicely complement the efforts of engineers to build and control swarms of robots and AI systems, but who as yet largely work with passive matter where competency exists only at one scale.

Taken together, collective intelligence is an extremely exciting and interdisciplinary emerging field that spans from the most fundamental philosophical problems of the parts-whole relationship to advancing fundamental and applied discovery in a number of important subfields.

## Reporting summary
Further information on research design is available in the Nature Portfolio Reporting Summary linked to this article.

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

## Acknowledgements

We thank Douglas Blackiston, Randall Ellis, and Patrick Erickson for their helpful comments on the manuscript, as well as Julia Poirier for editorial assistance. M.L. gratefully acknowledges support of the Guy Foundation Family Trust (103733-00001), of grants 62212 and 62230 from the John Templeton Foundation (the opinions expressed in this publication are those of the author(s) and do not necessarily reflect the views of the John Templeton Foundation), of the Templeton World Charity Foundation (TWCF0606) and of the Air Force Office of Scientific Research under award number FA9550-22-1-0465, Cognitive & Computational Neuroscience program.

## Author contributions

M.L. and P.M. wrote this Perspective together, including working on the text and creating the figures.

## Competing interests

The authors declare the following competing interests: Tufts University has a sponsored research agreement with a company, Astonishing Labs, to fund projects relevant to the collective intelligence of cells.
