## [Peer Review File · Communications Biology]

Reviewers' comments:

Reviewer #1 (Remarks to the Author):

The manuscript review revisits themes from earlier articles by the authors, and further develops the concept of collective intelligence as a "unifying concept" for biology. A number of examples of the competency of biological collectives to solve problems in diverse spaces are worked through in support of this thesis. By promising a synthetic perspective on hierarchically nested complex systems in biology, collective intelligence purportedly offers a global framework within which tools from formerly siloed disciplines - e.g. behavioural psychology and developmental biology - can be simultaneously drawn upon. Such transdisciplinary methodology may pave the way to breakthroughs in apparently diverse areas of medicine including psychiatry and regenerative medicine. Indeed, under such a view psychiatry itself may be conceptualised as a branch of regenerative medicine. Whilst I'm largely convinced of the promise of this framework, I have a number of sympathetic comments to make about the peril of excessive generalisation (or attempts at "unification").

One such comment concerns the utility of William James' definition as a unifying concept for "intelligence". A generic problem with definitions in biology (and maybe elsewhere, cf. Wittgenstein) is the necessity of the "I know it when I see it", clause. In my own disciplines of toxinology and pharmacology we have this challenge with concepts as fundamental as "venom" and "toxin". Due to the context-dependency of toxicity as well as the ecological role implied by the designation of "venom" as a function class, endless ad-hoc caveats become necessary to avoid concept creep. I won't bore the editors and authors with the specific details in those cases, but there is the similar possibility that if James' definition of intelligence as "a fixed goal with variable means of achieving it" is uncritically accepted, it becomes difficult to avoid the conclusion that any process constrained by the laws of physics is "intelligent". For example, if we roll 1000 boulders down a steep incline, each may chart a unique trajectory, but all will do as gravity dictates and come to rest at a local energy minimum. The logic might be maximised to certain arguments about the eventual "heat death" of the universe. Even allowing for degrees of freedom - for a non-Laplacean universe which may traverse any number of specific pathways - if all possible histories of the universe end in its inevitable heat death (I say nothing here of whether or not such an hypothesis is supported), might we not be forced to consider, under James' definition, the universe itself "intelligent"?

Perhaps this is not an issue for the authors, but might "intelligence" not cease to be a useful concept if we allow it to creep all the way to the horizon? Perhaps we can avoid this with a nuanced definition of "goal" - boulders and universes perhaps do not have "goals", which are properly assigned only to such agential systems as we find in biology. But "goal" may turn out to be equally as difficult to define as "intelligence", and "life" itself may appear to be in jeopardy if we start to push on such notions. This is why the tide of concept creep is so frequently (yet tacitly) halted with the "I know it when I see it" clause. However, the authors intend to *extend* our usual intuitions about intelligence by affirming the intelligence - or nanointentionality? - of cellular collectives or indeed collectives of sub-cellular components (i.e. individual cells). Thus they wish to explicitly advocate for a definition of "intelligence" and demonstrate its broader application than the intuition of biologists has hitherto admitted - they wish to question the tacit "I know it when I see it" clause. I'm personally very sympathetic to such extensions, and only wish to point out that bringing in James' pithy definition to support them may be somewhat facile. Closer attention to the *boundaries* of the region of dynamic reality delineated by the definition of "intelligence" may be warranted.

This may indeed represent one of the key challenges with the free energy principle more broadly - in its attempt to provide a unifying framework for life the universe and everything, it risks blurring boundaries that we may yet wish to preserve. And this may be the philosophical challenge of "unifying concepts" in general - the world appears exceptionally differentiated, yet apparently also unified at some level of description. How can we do justice in thought, in theory, in the usage of language, to this fundamental co-presence of the diverse and the unified? It's an ancient question, and one that seems like it might be with us a while longer.

Further, when we speak of the neural crest (or any other bio-system) achieving its morphogenetic goals despite novel circumstances, are we not speaking of *robustness* to perturbation? We may find this sort of robustness in bio-systems of all sorts. For example, due to redundancies (diverse toxins which fulfil the same trait-level function of e.g. prey subjugation) a venom system may remain capable of fulfilling its function despite various perturbations, including endogenous ones (shifts in gene expression, mutational events, etc.) and exogenous ones (the evolution of resistance to a particular toxin in a target organism). Are venom systems thus "intelligent", or just (functionally) "robust"? Might it not be the case that "intelligence" - or flexible cognitive problem solving - is a particular example of "robustness" rather than itself the generic case? If so, we can still have our cake and eat it - we can say that the intelligent systems studied by (e.g.) behavioural psychology are particularly robust (perhaps because particularly flexible/plastic?) and thus the tools we have used to study them are broadly applicable to the study of robustness in bio-systems at various levels. Thus we might avoid some of the infelicitous implications of excessively generalising "intelligence"? Indeed, in line 333 this parallel between robustness and intelligence is noted, but no discussion is provided of how we might disambiguate these two concepts - are we to consider them the same? I do hope not.

There is an awful lot more to say here about generalised evolutionary thinking and the capacity of evolving (or developing) systems to take (and in some cases re-take) *habits* or (meta)stable configurations, i.e. their capacity to be robust (is not robustness a form of stability?). I look forward to further engagement with the authors on these topics, but will avoid turning this review into an essay.

In any case, I do not think such philosophical considerations detract from the very real potential of these frameworks, so I will move on to more specific comments.

Line 94 - perhaps it should read "...centralized ones *possessed* by familiar animals..." unless the intention is to say that we animals *are* minds?

Line 95 - it's interesting that they say that neurons do not move relative to one another - presumably this refers only to the soma, as the dendrites and axons *do* move.

Line 116 - I seem to have definitively lost this particular terminological battle, but I still prefer "plesiotypic" or "plesiomorphic" to "basal" (which is a topological term derived from phylogenetic trees).

Line 127 - "explore" not "in exploring" (grammar)

Line 359 - "segmentation" appears twice. An example of redundancy - redundancy being one of the main mechanisms via which bio-systems achieve robustness, of course ;)

Line 409 - it is stated that all intelligences appear to be made of parts. Fair enough, but what entities in the known universe are *not* made of parts? And, if the parts themselves are "intelligent" (or conscious, or whatever other capacity we might be generalising - cf. notions of the principle of least action, or the path-integral, as "problem-solving"), then in what sense can we say that intelligence is intrinsically the property of collectives? Unless we add a reasoned cut off point to our definition of "intelligence", it seems (again) that we will have simply made every thing - simple or composite - in the universe "intelligent". Given the fundamental role of language in *differentiating* our understanding, this seems an undesirable consequence.

Line 417 - it's fascinating here to see the notion that components have continuous states/functions whereas the states of collectives - because of their goal-oriented or "finalist" (cf. Raymond Ruyer) natures - have discrete functions. This is a "strange inversion" of much thinking on emergence in which components are often considered to be "determined", but collectives are "more than the sum of their parts", i.e. have degrees of freedom the components do not possess. I'm sympathetic to this inversion, not least for the way it troubles lazy emergentist thinking. I'm reminded of Timothy Morton's notion of "subscendence" (as opposed, of course, to transcendence) and of a thought experiment I've encouraged many students to engage in viz. the relative ease of making a computer model of a flock of starlings - a murmuration - versus a model of the behaviour of a

single startling, the latter of which is vastly more unpredictable.

In conclusion, I find a lot to value and admire in this present paper, though I have devoted my time to a few key philosophical considerations. I do find the overall framework as exciting as ever. I certainly have no hesitation in recommending its publication.

Regards,

Timothy N. W. Jackson

Reviewer #2 (Remarks to the Author):

This exciting paper introduces a unifying framework to address the nature of cognition across scales grounded in collective intelligence. I like how the authors present this rich landscape, from cells to embryos.

As pointed out by the authors, collective intelligence has been traditionally used to describe the system-level dynamics of swarms of interacting agents such as ants, bees, flocks of birds, fish schools or their robotic counterparts. However, the conceptual framework of collective responses is required to sense, respond to, and adapt to external and internal signals that can be generalized to tissues and embryos.

This idea is backed up by many particularly compelling examples within the context of embryos as collective systems. This is a significant departure from the gene-based or cell-based views of morphogenesis and a powerful illustration of the relevance of collective dynamics beyond standard pattern formation ideas.

I believe this will be a very useful and insightful paper for those interested in the interconnections between pattern formation/morphogenesis/organogenesis and those processes requiring information processing. At the core of these connections, there is a picture of biological complexity that is still emerging where attractor dynamics in multicellular systems reveals at least two layers of explanation. The first is the morphogenetic one (what is been generated through developmental and regeneration processes) while the second is the cognitive one (where information and computation matter). The work presented here reveals that classic (and more recent) examples of developmental dynamics cannot be fully understood without the second layer. This is a very important insight and, as the authors show in their paper, a necessary one in the future scenarios of tissue and organ engineering.

I am not so sure about the use of "intelligence" in all cases. Developmental complexity surely deserves to be seen in terms of swarms and collective information processing, but intelligence suggests a more complex level of processing (where memory and learning play a role) that might not apply everywhere. That said, I am pretty aware of the problems of defining intelligence beyond decision making. Perhaps future work might consider the potential classes of information dynamics that is associated to the diverse set of examples discussed here.

I have nothing else to say beyond the previous comments. I just would point that, although the list of references is very complete and informative, a few items should be added:

(1) There is an early attempt to introduce the concept of a morphospace for organs and organoids that considers collective intelligence (ant colonies) as part of the whole picture and deserves to be cited: Oller, A. et al. 2016. A morphospace for synthetic organs and organoids: the possible and the actual. *Integrative Biology* 8, 485-503.

(2) One class of organisms that is too often forbidden when discussing cognition in biology

concerns plants. This is not the case here (and in previous papers by ML and co-workers) since references to plant roots (one of the critical examples of suggested embodiment for plant cognition) are cited. There is one very recent paper (which appeared online after the submission of this work) that deserves to be included since it explicitly addresses collective behaviour in plant tissues in different contexts, such as germination:

Davis, G.V. et al. 2023. Toward uncovering an operating system in plant organs. *Trends in Plant Science*. DOI:<https://doi.org/10.1016/j.tplants.2023.11.006>

Reviewer #1 (Remarks to the Author):

The manuscript review revisits themes from earlier articles by the authors, and further develops the concept of collective intelligence as a "unifying concept" for biology. A number of examples of the competency of biological collectives to solve problems in diverse spaces are worked through in support of this thesis. By promising a synthetic perspective on hierarchically nested complex systems in biology, collective intelligence purportedly offers a global framework within which tools from formerly siloed disciplines - e.g. behavioural psychology and developmental biology - can be simultaneously drawn upon. Such transdisciplinary methodology may pave the way to breakthroughs in apparently diverse areas of medicine including psychiatry and regenerative medicine. Indeed, under such a view psychiatry itself may be conceptualised as a branch of regenerative medicine. Whilst I'm largely convinced of the promise of this framework, I have a number of sympathetic comments to make about the peril of excessive generalisation (or attempts at "unification").

We thank the reviewer for their positive and constructive comments, and address all of the points as follows:

One such comment concerns the utility of William James' definition as a unifying concept for "intelligence". A generic problem with definitions in biology (and maybe elsewhere, cf. Wittgenstein) is the necessity of the "I know it when I see it", clause. In my own disciplines of toxinology and pharmacology we have this challenge with concepts as fundamental as "venom" and "toxin". Due to the context-dependency of toxicity as well as the ecological role implied by the designation of "venom" as a function class, endless ad-hoc caveats become necessary to avoid concept creep. I won't bore the editors and authors with the specific details in those cases, but there is the similar possibility that if James' definition of intelligence as "a fixed goal with variable means of achieving it" is uncritically accepted, it becomes difficult to avoid the conclusion that any process constrained by the laws of physics is "intelligent". For example, if we roll 1000 boulders down a steep incline, each may chart a unique trajectory, but all will do as gravity dictates and come to rest at a local energy minimum. The logic might be maximised to certain arguments about the eventual "heat death" of the universe. Even allowing for degrees of freedom - for a non-Laplacean universe which may traverse any number of specific pathways - if all possible histories of the universe end in its inevitable heat death (I say nothing here of whether or not such an hypothesis is supported), might we not be forced to consider, under James' definition, the universe itself "intelligent"?

This is an important point and we have now added a Box to the manuscript addressing this issue. The TAME framework which we have developed for this purpose says several things that are relevant to the above. First, that no claims about intelligence can be made from purely observational data. That is, we argue that claims that a cell, a plant, an ecosystem, or the Universe *is* intelligent (as made by animist prescientific societies) are as wrong as claims that cells, plants, ecosystems, and the Universe *can't be* intelligent (as often claimed by scientists). The problem is that this is an empirical matter, and cannot be decided based on philosophical commitments or *a priori* decisions. Our view does not entail the Universe, or balls rolling down hills, to be or not be intelligent until specific experiments are done. James gives us the beginnings of the research roadmap, and we

have added more in recent work. What we need to do is attempt the tools of behavioral science – put barriers in its place, train it using various paradigms that reveal different types of learning, etc. and then we find out whether those tools offer any improved prediction and control. We know of no way to make this determination for the Universe, but for example, in the case of the weather, which people often bring up as an obvious case where one wouldn't find intelligence, we actually could find out – for all we know, it may have aspects of habituation, sensitization, and who knows what else, if properly stimulated and studied. A specific case concerns gene regulatory networks, which everyone thought were mechanical and stupid (because they are transparent, deterministic, and simple); once we checked [1, 2], we found evidence of 6 different kinds of learning just in those networks alone, without the rest of cellular machinery or synapses. We have not yet found evidence of anticipation, planning, or high-order cognitive properties, so our claim would be of modest, yet non-zero, cognitive capacity.

This leads us to the second point, terminological creep. We suggest that while it cannot (and probably should not) be entirely avoided (especially with novel systems that often stretch definitions crafted for a specific set of statements), it can be kept in check by adhering to a very practical criterion. The criterion shouldn't be "I know it when I see it", but rather, "I know it when it helps do new things that haven't been done before". That is, we can support the claim that a category (e.g., problem-solving intelligence) applies to a given model, if and only if, we can demonstrate that by using that framing, we gain the ability to discover new things and reach capabilities that were not available before. In other words, if it helps progress. We note parenthetically that this does not rule out the fact that *after* someone shows a new capability, past framings can often be used (epicycle-like) to explain why it makes sense and doesn't destroy the other paradigm. The real question, we submit, is what novel discoveries are facilitated by a given choice of definition. The goal of our perspective piece is to highlight for readers what the gains might be of exploring this framing, and we cite a number of papers in which we have described the new research that was uniquely facilitated by this unusual way of interpreting certain biological observations.

Perhaps this is not an issue for the authors, but might "intelligence" not cease to be a useful concept if we allow it to creep all the way to the horizon? Perhaps we can avoid this with a nuanced definition of "goal" - boulders and universes perhaps do not have "goals", which are properly assigned only to such agential systems as we find in biology.

We are sympathetic to this, but in defense of boulders, I note that Least Action principles are extremely powerful and apply usefully (in the engineering sense) to boulders, photon paths, and a lot of other things. We have argued elsewhere that if one takes the spectrum of intelligence seriously, and asks what the absolute lowest, simplest, basal form of it would look like, it would look precisely like the minimal ability of many systems to navigate space in a way that is described by Least Action dynamics.

But "goal" may turn out to be equally as difficult to define as "intelligence", and "life" itself may appear to be in jeopardy if we start to push on such notions. This is why the tide of concept creep is so frequently (yet tacitly) halted with the "I know it when I see it" clause. However, the authors

intend to *extend* our usual intuitions about intelligence by affirming the intelligence - or nanointentionality? - of cellular collectives or indeed collectives of sub-cellular components (i.e. individual cells). Thus they wish to explicitly advocate for a definition of "intelligence" and demonstrate its broader application than the intuition of biologists has hitherto admitted - they wish to question the tacit "I know it when I see it" clause. I'm personally very sympathetic to such extensions, and only wish to point out that bringing in James' pithy definition to support them may be somewhat facile. Closer attention to the *boundaries* of the region of dynamic reality delineated by the definition of "intelligence" may be warranted.

We agree with the need for caution, and have emphasized in the new Box the need for empirical backing to those claims, since whether one "sees it" is otherwise completely dependent on that observer's background and agenda.

This may indeed represent one of the key challenges with the free energy principle more broadly - in its attempt to provide a unifying framework for life the universe and everything, it risks blurring boundaries that we may yet wish to preserve. And this may be the philosophical challenge of "unifying concepts" in general - the world appears exceptionally differentiated, yet apparently also unified at some level of description. How can we do justice in thought, in theory, in the usage of language, to this fundamental co-presence of the diverse and the unified? It's an ancient question, and one that seems like it might be with us a while longer.

Agreed; and indeed some of the most exciting aspects of the FEP center on the use of this principle to drive experiments that would otherwise not have been done. This is happening in neuroscience, also in our lab in the context of cell-scale preparations.

Further, when we speak of the neural crest (or any other bio-system) achieving its morphogenetic goals despite novel circumstances, are we not speaking of *robustness* to perturbation? We may find this sort of robustness in bio-systems of all sorts. For example, due to redundancies (diverse toxins which fulfil the same trait-level function of e.g. prey subjugation) a venom system may remain capable of fulfilling its function despite various perturbations, including endogenous ones (shifts in gene expression, mutational events, etc.) and exogenous ones (the evolution of resistance to a particular toxin in a target organism). Are venom systems thus "intelligent", or just (functionally) "robust"? Might it not be the case that "intelligence" - or flexible cognitive problem solving - is a particular example of "robustness" rather than itself the generic case? If so, we can still have our cake and eat it - we can say that the intelligent systems studied by (e.g.) behavioural psychology are particularly robust (perhaps because particularly flexible/plastic?) and thus the tools we have used to study them are broadly applicable to the study of robustness in bio-systems at various levels. Thus we might avoid some of the infelicitous implications of excessively generalising "intelligence"? Indeed, in line 333 this parallel between robustness and intelligence is noted, but no discussion is provided of how we might disambiguate these two concepts - are we to consider them the same? I do hope not.

Indeed, there are always systems that may be amenable to two (or more) frames. For example, a repair-person coming to fix the thermostat may say that they are a hard-nosed reductionist and don't believe in goals. They eschew cybernetics, and will instead stick to a purely physics-based model of the heating system. They may indeed succeed in fixing it, because a thermostat is such a simple goal-seeking system that it is - just barely

– still amenable to treatment like a simple machine. However, what they won't do is design the next, improved, heating system. Thus, some biological systems are well-described by "robustness" and that may indeed give us everything there is to be had, for that system. Some offer such a kind of competent robustness that progress requires us to abandon the robustness framing, and reach for useful concepts from the fields of navigational autonomous agents, planners, and other kinds of more complex problem-solvers. A (future) self-driving car that can get you to your destination despite all kinds of barriers (including maybe that the passenger forgot something important and it knows where to stop along the way, to make the whole trip worthwhile) cannot be usefully explained as "simply being robust with respect to rolling down to its destination", even though it obeyed the laws of physics the whole time. And, more importantly, in such a case, the conceptual tools afforded by robustness do not facilitate repair, design, or improvements on the system. We argue that biology occupies a very wide spectrum, containing systems that are well-handled by robustness/redundancy/degeneracy etc. and ones in which other disciplines offer better roadmaps for advances.

There is an awful lot more to say here about generalised evolutionary thinking and the capacity of evolving (or developing) systems to take (and in some cases re-take) *habits* or (meta)stable configurations, i.e. their capacity to be robust (is not robustness a form of stability?). I look forward to further engagement with the authors on these topics, but will avoid turning this review into an essay.

We look forward to it! This sounds like very fertile ground for a future paper.

In any case, I do not think such philosophical considerations detract from the very real potential of these frameworks, so I will move on to more specific comments.

Line 94 - perhaps it should read "...centralized ones *possessed* by familiar animals..." unless the intention is to that that we animals *are* minds?

Yes, absolutely correct, we have updated the text.

Line 95 - it's interesting that say that neurons do not move relative to one another - presumably this refers only to the soma, as the dendrites and axons *do* move.

True! We have qualified our statement.

Line 116 - I seem to have definitively lost this particular terminological battle, but I still prefer "plesiotypic" or "plesiomorphic" to "basal" (which is a topological term derived from phylogenetic trees).

We have made the change to "plesiomorphic".

Line 127 - "explore" not "in exploring" (grammar)

Fixed.

Line 359 - "segmentation" appears twice. An example of redundancy - redundancy being one the main mechanisms via which bio-systems achieve robustness, of course ;)

Fixed.

Line 409 - it is stated that all intelligences appear to be made of parts. Fair enough, but what entities in the known universe are **not** made of parts? And, if the parts themselves are "intelligent" (or conscious, or whatever other capacity we might be generalising - cf. notions of the principle of least action, or the path-integral, as "problem-solving"), then in what sense can we say that intelligence is intrinsically the property of collectives? Unless we add a reasoned cut off point to our definition of "intelligence", it seems (again) that we will have simply made every thing - simple or composite - in the universe "intelligent". Given the fundamental role of language in **differentiating** our understanding, this seems an undesirable consequence.

Good point. This refers to the *scaling* of intelligence: rocks have no more intelligence than their parts (if for example we believed that elementary particles, due to the least action principle, had some sort of non-zero femto-competencies). However, biology is great at scaling the intelligence of its parts into higher levels of intelligence and new problem spaces. In fact that is a potential definition of life – systems which are good at scaling their parts' agency (but we won't go into depth on that here, as it's beyond the scope of this paper). We address this in detail in several theoretical and computational primary papers [3-6] and have now added a brief reference to this in the text.

Line 417 - it's fascinating here to see the notion that components have continuous states/functions whereas the states of collectives - because of their goal-oriented or "finalist" (cf. Raymond Ruyer) natures - have discrete functions. This is a "strange inversion" of much thinking on emergence in which components are often considered to be "determined", but collectives are "more than the sum of their parts", i.e. have degrees of freedom the components do not possess. I'm sympathetic to this inversion, not least for the way it troubles lazy emergentist thinking. I'm reminded of Timothy Morton's notion of "subscendence" (as opposed, of course, to transcendence) and of a thought experiment I've encouraged many students to engage in viz. the relative ease of making a computer model of a flock of starlings - a murmuration - versus a model of the behaviour of a **single** starling, the latter of which is vastly more unpredictable.

These are fascinating ideas that expand our thesis, and we will follow up on them for the next paper.

In conclusion, I find a lot to value and admire in this present paper, though I have devoted my time to a few key philosophical considerations. I do find the overall framework as exciting as ever. I certainly have no hesitation in recommending its publication.

We once again thank the reviewer for the opportunity to engage productively in these fascinating issues. The paper has indeed been improved by the suggested modifications.

Reviewer #2 (Remarks to the Author):

This exciting paper introduces a unifying framework to address the nature of cognition across scales grounded in collective intelligence. I like how the authors present this rich landscape, from cells to embryos. As pointed out by the authors, collective intelligence has been traditionally used

to describe the system-level dynamics of swarms of interacting agents such as ants, bees, flocks of birds, fish schools or their robotic counterparts. However, the conceptual framework of collective responses is required to sense, respond to, and adapt to external and internal signals that can be generalized to tissues and embryos. This idea is backed up by many particularly compelling examples within the context of embryos as collective systems. This is a significant departure from the gene-based or cell-based views of morphogenesis and a powerful illustration of the relevance of collective dynamics beyond standard pattern formation ideas. I believe this will be a very useful and insightful paper for those interested in the interconnections between pattern formation/morphogenesis/organogenesis and those processes requiring information processing. At the core of these connections, there is a picture of biological complexity that is still emerging where attractor dynamics in multicellular systems reveals at least two layers of explanation. The first is the morphogenetic one (what is been generated through developmental and regeneration processes) while the second is the cognitive one (where information and computation matter). The work presented here reveals that classic (and more recent) examples of developmental dynamics cannot be fully understood without the second layer. This is a very important insight and, as the authors show in their paper, a necessary one in the future scenarios of tissue and organ engineering.

We thank the reviewer for their helpful comments, and have addressed them as follows:

I am not so sure about the use of "intelligence" in all cases. Developmental complexity surely deserves to be seen in terms of swarms and collective information processing, but intelligence suggests a more complex level of processing (where memory and learning play a role) that might not apply everywhere. That said, I am pretty aware of the problems of defining intelligence beyond decision making. Perhaps future work might consider the potential classes of information dynamics that is associated to the diverse set of examples discussed here.

We agree, and have now added a short text to clarify that 1) by intelligence, we do not simply mean emergent complexity, but *some degree of* competency to solve problems and get a task accomplished in novel circumstances; and 2) some of these processes do indeed involve memory and learning (for example, such as that seen in gene regulatory network models [1, 2]), but intelligence is a broad spectrum that includes simpler versions of problem-solving as well.

I have nothing else to say beyond the previous comments. I just would point that, although the list of references is very complete and informative, a few items should be added: (1) There is an early attempt to introduce the concept of a morphospace for organs and organoids that considers collective intelligence (ant colonies) as part of the whole picture and deserves to be cited: Oller, A. et al. 2016. A morphospace for synthetic organs and organoids: the possible and the actual. *Integrative Biology* 8, 485-503.

Excellent point, we have added this very nice paper.

(2) One class of organisms that is too often forbidden when discussing cognition in biology concerns plants. This is not the case here (and in previous papers by ML and co-workers) since references to plant roots (one of the critical examples of suggested embodiment for plant

cognition) are cited. There is one very recent paper (which appeared online after the submission of this work) that deserves to be included since it explicitly addresses collective behaviour in plant tissues in different contexts, such as germination: Davis, G.V. et al. 2023. Toward uncovering an operating system in plant organs. Trends in Plant Science. DOI:<https://doi.org/10.1016/j.tplants.2023.11.006>

We had been unaware of this fascinating paper, and now cite it.

References Cited in Response

1. Biswas, S., W. Clawson, and M. Levin, *Learning in Transcriptional Network Models: Computational Discovery of Pathway-Level Memory and Effective Interventions*. Int J Mol Sci, 2022. **24**(1).
2. Biswas, S., et al., *Gene Regulatory Networks Exhibit Several Kinds of Memory: Quantification of Memory in Biological and Random Transcriptional Networks*. iScience, 2021. **24**(3): p. 102131.
3. Levin, M., *The Computational Boundary of a "Self": Developmental Bioelectricity Drives Multicellularity and Scale-Free Cognition*. Frontiers in Psychology, 2019. **10**(2688): p. 2688.
4. Pio-Lopez, L., et al., *The scaling of goals from cellular to anatomical homeostasis: an evolutionary simulation, experiment and analysis*. Interface Focus, 2023. **13**(3): p. 20220072.
5. Levin, M., *Technological Approach to Mind Everywhere: An Experimentally-Grounded Framework for Understanding Diverse Bodies and Minds*. Frontiers in Systems Neuroscience, 2022. **16**: p. 768201.
6. Fields, C. and M. Levin, *Competency in Navigating Arbitrary Spaces as an Invariant for Analyzing Cognition in Diverse Embodiments*. Entropy (Basel), 2022. **24**(6).

REVIEWERS' COMMENTS:

Reviewer #1 (Remarks to the Author):

Thanks for sharing this – I've enjoyed reading the author's responses to reviewer queries.